# Blood and tissue biomarker analysis in dogs with osteosarcoma treated with palliative radiation and intra-tumoral autologous natural killer cell transfer

Sean J. Judge[1]☯, Mio Yanagisawa[1]☯, Ian R. Sturgill[1]☯, Sarah B. Bateni[1], Alicia A. Gingrich[1], Jennifer A. Foltz[2], Dean A. Lee[2], Jaime F. Modiano[3], Arta M. Monjazeb[4], William T. N. Culp[5], Robert B. Rebhun[5], William J. Murphy[6], Michael S. Kent[5‡], Robert J. Canter[7‡]*

1 Department of Surgery, University of California Davis Medical Center, Sacramento, California, United States of America, 2 Nationwide Children's Hospital, Center for Childhood Cancer & Blood Diseases, Columbus, Ohio, United States of America, 3 Department of Veterinary Clinical Sciences, College of Veterinary Medicine, Animal Cancer Care and Research Center, Center for Immunology, Masonic Cancer Center, and Stem Cell Institute, University of Minnesota, St. Paul, Minneapolis, United States of America, 4 Department of Radiation Oncology, University of California Davis Medical Center, Sacramento, California, United States of America, 5 The Center for Companion Animal Health, Department of Surgical and Radiological Sciences, School of Veterinary Medicine, University of California Davis, Davis, California, United States of America, 6 Distinguished Professor of Dermatology and Internal Medicine, Vice Chair of Dermatology, University of California Davis Medical Center, Sacramento, California, United States of America, 7 Department of Surgery, Division of Surgical Oncology, University of California Davis Medical Center, Sacramento, California, United States of America

☯ These authors contributed equally to this work.
‡ These authors also contributed equally to this work.
* rjcanter@ucdavis.edu

**Data Availability Statement:** All relevant data are within the manuscript and its Supporting Information files.

## Abstract

We have previously reported radiation-induced sensitization of canine osteosarcoma (OSA) to natural killer (NK) therapy, including results from a first-in-dog clinical trial. Here, we report correlative analyses of blood and tissue specimens for signals of immune activation in trial subjects. Among 10 dogs treated with palliative radiotherapy (RT) and intra-tumoral adoptive NK transfer, we performed ELISA on serum cytokines, flow cytometry for immune phenotype of PBMCs, and PCR on tumor tissue for immune-related gene expression. We then queried The Cancer Genome Atlas (TCGA) to evaluate the association of cytotoxic/immune-related gene expression with human sarcoma survival. Updated survival analysis revealed five 6-month survivors, including one dog who lived 17.9 months. Using feeder line co-culture for NK expansion, we observed maximal activation of dog NK cells on day 17–19 post isolation with near 100% expression of granzyme B and NKp46 and high cytotoxic function in the injected NK product. Among dogs on trial, we observed a trend for higher baseline serum IL-6 to predict worse lung metastasis-free and overall survival (P = 0.08). PCR analysis revealed low absolute gene expression of CD3, CD8, and NKG2D in untreated OSA. Among treated dogs, there was marked heterogeneity in the expression of immune-related genes pre- and post-treatment, but increases in CD3 and CD8 gene expression were higher among dogs that lived > 6 months compared to those who did not. Analysis of the TCGA

**Funding:** This work was supported in part by National Institute for Health/National Cancer Institute grant R01 CA189209 (WJM) and U01 CA224166–01 (RJC, RBR). Additional funding was provided by the Society for Surgical Oncology Foundation (RJC), the Sarcoma Foundation of America (RJC), and the University of California Coordinating Committee for Cancer Control CRR-13-201,404 (RJC).

**Competing interests:** The authors have declared that no competing interests exist.

confirmed significant differences in survival among human sarcoma patients with high and low expression of genes associated with greater immune activation and cytotoxicity (CD3e, CD8a, IFN-γ, perforin, and CD122/IL-2 receptor beta). Updated results from a first-in-dog clinical trial of palliative RT and autologous NK cell immunotherapy for OSA illustrate the translational relevance of companion dogs for novel cancer therapies. Similar to human studies, analyses of immune markers from canine serum, PBMCs, and tumor tissue are feasible and provide insight into potential biomarkers of response and resistance.

## Introduction

Although approximately 60–70% of human patients with osteosarcoma (OSA) treated with standard of care multimodality therapy experience long-term survival, these numbers have remained static for the past three decades, and the survival of patients with relapsed and/or metastatic OSA is dismal [1–3]. Despite evidence that OSA can be recognized by the immune system [4–6], the clinical experience with immunotherapy for OSA has been disappointing. Moreover, evaluating and optimizing immunotherapy for an orphan disease where up to 70% of patients experience long-term survival is challenging. Indeed, poor compliance and retention in this patient population has been reported in the recently completed worldwide EURA-MOS trial [7], clearly demonstrating the challenges of evaluating immunotherapy in a disease with overall favorable, but variable outcomes.

Companion dog OSA demonstrates remarkable similarity to the complexity and heterogeneity of human OSA [8,9]. Importantly, unlike murine models of cancer based on tumor injections and/or genetic-engineering, dogs develop spontaneous cancers in the setting of an intact immune system, and naturally-occurring cancers in dogs recapitulate the important host/tumor principles of immune equilibrium, immune evasion, and immune escape [8,10,11]. In addition, the burden of dog cancer, including OSA, is significant, and some investigators have estimated that the prevalence of cancer in companion dogs is higher than that of humans [8]. Since there is a less established standard of care in veterinary medicine, investigational treatments, such as immunotherapy, can be considered earlier in the course of therapy, thereby allowing for testing of novel therapies that can speed translation of innovative therapies to both dogs and humans [11]. Taken together, dogs with naturally occurring cancer represent an ideal model to evaluate novel immunotherapy approaches. Trials in companion animals are an important bridge between pre-clinical testing in murine models and clinical trials in people and allow for important proof-in-concept studies of innovative new strategies. Concurrently, dog clinical trials provide key preliminary data on efficacy and toxicity in a large animal model with notable similarity to people [11]. However, canine trials remain nascent, and more studies are needed to underscore the generalizability of the findings, the similarities of the immune readouts, and the validation of key reagents.

Given our preliminary data showing that radiotherapy (RT) sensitizes tumors, including sarcomas, to NK cytotoxicity as well as the unmet need for effective immunotherapy in OSA [12], we previously conducted a clinical trial in dogs of palliative RT plus intra-tumoral autologous NK transfer in dogs with non-metastatic OSA whose owners elected not to pursue amputation or cytotoxic chemotherapy [13]. As part of this clinical trial, we collected serial blood and tumor specimens pre- and post-treatment to assess serum cytokines, to evaluate immune phenotype of circulating PBMCs, and to analyze gene expression in tumor tissue. Our objective was to analyze blood and tissue specimens as correlates of clinical response to NK/RT

immunotherapy, and we hypothesized that analysis of blood and tumor tissue would further validate the dog model for canine immunotherapy studies and would provide preliminary evidence of local or systemic immune response to treatment for testing in a larger trial. As our clinical trial enrollment in dogs was small, we also queried The Cancer Genome Atlas (TCGA) to analyze the association of intra-tumoral immune gene expression with survival in human sarcomas to further inform the relevance of dog sarcomas in the evaluation of novel immunotherapy approaches on the path to human clinical testing.

## Materials and methods

### First-in-Dog clinical trial

The schema of the enrollment, NK isolation and expansion, and radio-immunotherapy protocol have been described previously [13]. In brief, dogs were considered eligible if they were diagnosed with locally advanced, non-metastatic OSA, had adequate end organ function, and were not pursuing amputation or chemotherapy. Our study enrollment was designed to accrue 10 patients in order to evaluate the primary endpoint of lung metastasis formation at 6 months. The clinical trial was approved by the UC Davis School of Veterinary Medicine Clinical Trials Review Board and the Institutional Animal Care and Use Committee (protocol #18857).

The clinical protocol consisted of 4 fractions of palliative RT at a dose of 9 Gy weekly followed by two intra-tumoral injections of *ex vivo* expanded autologous canine NK cells (7.5 x $10^6$ NK cells/kg) in 1–2 mL aliquots in sterile PBS during weeks 5 and 6 [13]. Ex vivo expanded canine NK cells are obtained similar to the method previously described [14,15], utilizing co-culture of canine peripheral blood with irradiated human erythroleukemia K562 cell line engineered to express membrane bound IL-21 and 4-1bb ligand, supplemented with 100 IU/mL rhIL-2. Recombinant human IL-2 was co-injected with the NK cells at a dose of 250,000 IU/kg (Roche, NCI, Frederick). Resting dog NK cells were isolated from a CD5$^{dim}$ population [16], but when activated and mature were phenotyped as CD5-/CD3-/NKp46+ [17]. We assessed for lung metastasis using thoracic computed tomography every three months for the first six months of follow-up followed by thoracic radiographs every three months thereafter.

### Serum cytokine measurement

Pre-treatment serum cytokine levels were measured as a baseline using serum isolated from canine peripheral blood by centrifugation. Serum concentrations of IL-2, IL-6, and TNFα were measured using dog-specific ELISA plates per manufacturer's specifications (Kingfisher Biotech, MN, USA). Pre-treatment values were retrospectively correlated with survival.

### qRT-PCR

Total RNA was extracted from PBMCs and CD5-depleted cells using the RNeasy Mini Kit (Qiagen) and from tumor biopsies and normal lymph node using the RNeasy Fibrous Tissue Mini Kit (Qiagen). Tumor samples were first stored in RNAlater solution (Invitrogen) before being homogenized using a TissueLyser II system (Qiagen). Extracted RNA was reverse transcribed to cDNA using iScript Reverse Transcription Supermix (Bio-Rad). Gene-specific primers were obtained from Bio-Rad and Integrated DNA Technologies; additional primer information can be found in S1 Table. Quantitative real-time PCR was performed using SsoAdvanced Universal SYBR Green Supermix (Bio-Rad) and the StepOnePlus Real-Time PCR system (Applied Biosystems). Briefly, reactions containing 5 ng of cDNA were subjected to the thermal cycling conditions recommended by the SsoAdvanced manual for use with the

StepOnePlus system: initial 30 seconds at 95 degrees Celsius followed by 40 cycles of denaturation for 15 seconds at 95 degrees Celsius and annealing and extension for 60 seconds at 60 degrees Celsius.

## Flow cytometry

$5 \times 10^5–10^6$ cells were stained in round bottom 96 well plates. Surface antibodies were diluted with staining buffer (2% FBS, 1 mM EDTA, and 0.02% NaN3 in PBS) and blocking buffer using canine Fc receptor binding inhibitor (ThermoFisher, #14–9162–42) and canine gamma globulin (Jackson ImmunoResearch, #004–000–002). Details regarding specific antibodies utilized in this study can be found in S2 Table. For intracellular staining of canine interferon-γ and granzyme B, cells were mixed with viability stain, then washed and incubated with fixation and permeabilization solution per manufacturer's instructions (BD Biosciences). We then incubated with intracellular stain or isotype prepared in Perm/Wash Buffer™ followed by centrifugation and resuspension in 1% paraformaldehyde for flow cytometry analysis. Fluorescent minus one (FMO) controls containing all the fluorochromes in the panel except the one for that marker of interest were used to control for fluorescent spread due to multiple fluorochromes in a given panel.

## Cytotoxicity assays

NK cytotoxicity was determined by flow cytometry using co-culture assays with expanded NK cells and the prototypical dog NK target line, canine thyroid adenocarcinoma cells (CTAC) [13,16]. SSC$^{hi}$ CD45-7AAD– and SSChi CD45-7AAD+ populations were analyzed relative to control tumor cells not exposed to NK co-culture. All data were collected using a BD Fortessa flow cytometer equipped with BD FACSDiva software (BD Biosciences, San Jose, CA). Data were analyzed using FlowJo software (TreeStar, Ashland, OR). The characteristics of our purified mouse anti-canine NKp46 antibody have been described previously [17].

## Animal studies

As noted above, the clinical trial portion of this study was approved by the UC Davis School of Veterinary Medicine Clinical Trials Review Board and IACUC (protocol #18857) and consisted of 10 dogs with naturally occurring, spontaneous OSA. Additional client-owned dogs with OSA (N = 2) or soft tissue sarcoma (N = 3) underwent surgery at the UC Davis Health Veterinary Medical Teaching Hospital (VMTH), and tumor tissue was obtained with owner consent (protocol #18315) for immunohistochemistry and immune analysis. Lymph node tissue was obtained from a dog undergoing necropsy, also with owner consent (protocol #20416). Blood was also obtained from farm-bred beagles (N = 6) per commercial relationship (Ridglan Farms, Inc., Mt. Horeb, WI) using EDTA tubes diluted with sterile PBS. Per Ridglan literature, their beagle colony is maintained using the strictest standards of quality in breeding, socialization, and animal welfare.

## The cancer genome atlas

Using the Data Matrix from TCGA website (https://tcgadata.nci.nih.gov/tcga/dataAccessMatrix. htm), we downloaded clinical and genomic data from the TCGA provisional soft tissue sarcoma (STS) data set on May 30, 2018 using the TCGA data portal (https://portal.gdc.cancer.gov/). Gene expression data for CD3e, CD8a, IFNG, GZMB, PRF1, CD122, IL-6, and IL-6R (CD126) were downloaded from the Computational Biology Center at Memorial Sloan-Kettering cBio-Portal website (http://www.cbioportal.org/). Using TCGA barcodes, we matched the TCGA

clinical and genomic data for individual patients. We compared overall survival (OS) for high and low gene expression using the 1st and 4th quartiles of the TCGA genes of interest using the Kaplan-Meier method, the log-rank test, and Cox proportional hazards models.

## Statistical considerations

Summary statistics were reported as mean ± standard error with median (range) where appropriate. Categorical variables were compared using a chi-squared test. Parametric continuous variables were compared using an independent samples t-test. Non-parametric continuous variables were compared using the Mann-Whitney U test. For comparison of more than two groups, statistical significance was determined using a one-way ANOVA followed by a Bonferroni multiple-group comparison test. Survival curves were created using the Kaplan-Meier method. Statistical analyses were performed using SAS version 9.2 (SAS Institute Inc., Cary, NC) and Graph-Pad Prism 5. Significance was set at $P < 0.05$.

## Results

### First-in-dog trial of RT and NK cell immunotherapy

Clinical characteristics and vital status of the canine patients from the clinical trial are detailed in Table 1. Among the 10 treated dogs, the mean age was 8.0 ± 3.3 years, and four were female (40%). The median body condition score was 6 (range 3–8), and the mean weight was 50 ± 21 kg. In all, the trial included three St. Bernards, three mixed breed dogs, and one each of the following: Labrador Retriever, Doberman pinscher, Pyrenees, and Rhodesian Ridgeback. The majority of OSA were located at the distal radius (60%), with other sites being the ilium (20%) and humerus (20%). With a median follow up of 5.7 months (9.7 months for survivors), six patients died for an overall mortality of 60%. Four dogs were euthanized secondary to progressive OSA, while 2 patients died from causes unrelated to cancer (1 from a perforated ulcer presumably from non-steroidal anti-inflammatory drugs and 1 from euthanasia after suffering a pathologic fracture post RT and local recurrence). The updated event-free overall and progression-free survival for the cohort are shown in Fig 1.

Table 1. Clinical characteristics and vital status of canine patients on clinical trial.

| Characteristic | | Number (%) |
|---|---|---|
| Age, mean ± SD | | 8.0 ± 3.3 |
| Sex | Male | 6 (60%) |
| | Female | 4 (40%) |
| Body Condition Score median (range) | | 6 (3–8) |
| Weight, mean (kg) ± SD | | 50 ± 21 |
| Breed | St. Bernard | 3 (30%) |
| | Shepherd Mix | 2 (20%) |
| | Other* | 5 (50%) |
| Tumor Location | Radius | 6 (60%) |
| | Ilium | 2 (20%) |
| | Humerus | 2 (20%) |
| Vital Status | Alive | 4 (40%) |
| | Dead | 6 (60%) |

* Includes Black Lab, Pyrenees, Doberman, Retriever Mix, and Rhodesian Ridgeback.

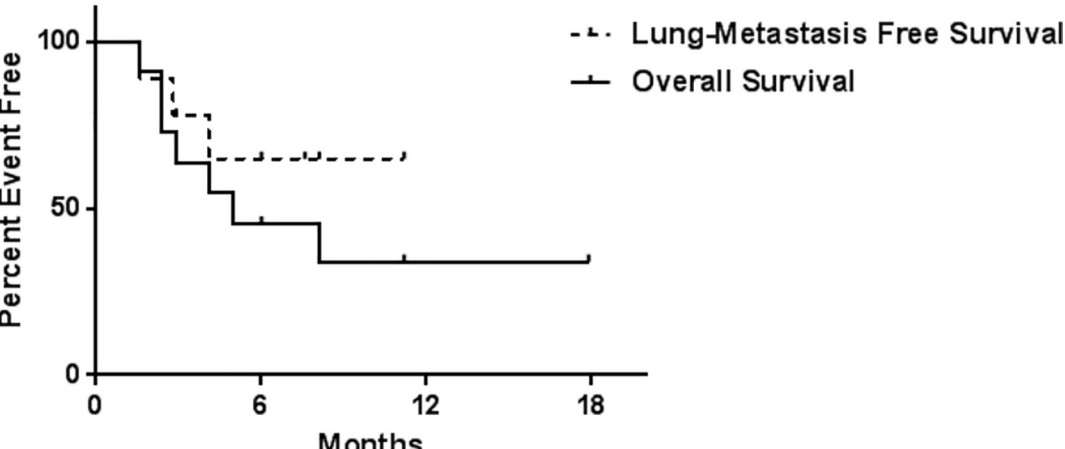

**Fig 1. Kaplan-Meier analysis of updated event-free and overall survival for ten dogs on trial.** Kaplan-Meier survival curves for lung-metastasis free survival (dashed line) and overall survival (solid line) for dogs with locally advanced, non-metastatic osteosarcoma treated with palliative radiotherapy and intra-tumoral NK transfer x2. Median survival was not reached for lung-metastasis free survival during the follow up period.

### Ex Vivo NK cell expansion

PBMCs were isolated from trial subjects, and NK cells were isolated for activation and expansion using CD5 depletion to enrich for the CD5$^{dim}$ subset. As shown in Fig 2A, pre-depletion PBMCs from trial subjects and healthy laboratory beagles are characterized by absent NKp46 expression and mixed CD3 expression, whereas the CD5 depleted population is enriched for NKp46+ cells which are CD3 negative. After two weeks in co-culture with an irradiated feeder line [18], the CD3-NKp46+ population markedly expands from $0.6 \pm 0.5\%$ in circulating PBMCs to $8.1 \pm 2.7\%$ at day 0 post CD5 depletion to $81.3 \pm 13.9\%$ at day 14 in co-culture ($P < 0.05$ from N = 6 beagles). As shown in Fig 2B, there is concomitant upregulation of cytotoxic marker granzyme B after CD5 depleted cells are co-cultured in K562.clone9 cells for 14 days with 100 IU/mL rhIL-2.

We also assessed NK cytotoxic function in killing assays from selected time points using both trial subjects and laboratory beagles. As shown in Fig 2C, after 17–20 days in co-culture NK cells demonstrated maximal cytotoxicity to canine thyroid adenocarcinoma (CTAC) target cells although we observed no statistically significant differences in cytotoxic function between day 17–20 and day 14 NK cells ($P > 0.05$) using blood from 3 beagles. However, by day 20–24 post expansion, we observed a diminution in NK cytotoxic function which was significantly less than day 14 and day 17–20 cells (although elevated above resting PBMCs, $P < 0.01$).

### Serum cytokines

Fig 3 highlights the baseline/ pre-treatment serum cytokine levels between dogs who lived $\geq 6$ months and dogs who died within 6 months of study entry. Mean canine serum IL-2 by ELISA (Fig 3A) was $99.4 \pm 53.4$ pg/ mL in 6-month survivors (our primary endpoint for the trial) compared to $162.9 \pm 124.9$ ($P > 0.05$) in non-survivors. Mean canine serum TNF-α (Fig 3C) was $42.5 \pm 60.1$ pg/ mL in 6-month survivors compared to $44.9 \pm 44.9$ in non-survivors ($P > 0.05$). As shown in Fig 3B, the difference in serum IL-6 between dogs who lived $\geq 6$ months ($0.0 \pm 0$ pg/ mL) and those who died within 6 months ($32.1 \pm 19.2$ pg/ mL) was not

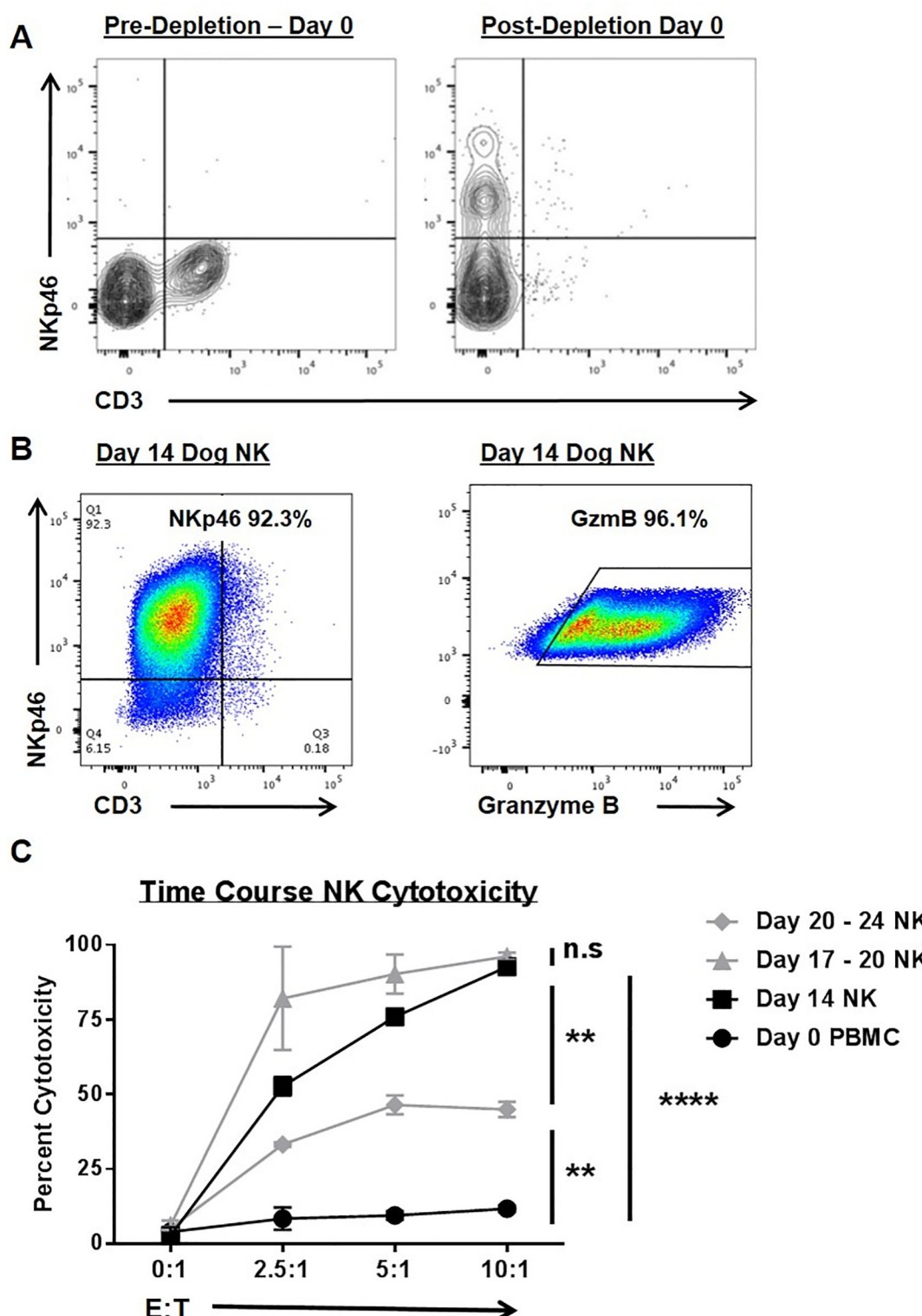

**Fig 2. Canine NK cells are enriched in NKp46 and Granzyme B expression with time-dependent changes in cytotoxic function.** (A) Canine NK cells were isolated from peripheral blood mononuclear cells via CD5 antibody depletion (left), thereby enriching for a CD3-NKp46+ population of cells (right). (B) Ex vivo expansion with an irradiated feeder cell line (human K562 leukemia line transduced with 4-1BBL and membrane bound IL-21) yields a CD3 population of cells that are markedly positive for NKp46+ (left) and Granzyme B+ (right) after 14 days of stimulation. For panels A and B, representative

flow cytometry plots from 10 study dogs and 6 healthy beagles are shown. (C) The cytotoxicity of NK cells was assayed at multiple time points (using CTAC cells as targets) and compared to fresh PBMCs (incubated with 100 IU/mL rhIL-2) in 12–16 hour killing assays. Data from one experiment performed in triplicate are shown. Mean values ± SD are shown. This experiment was repeated with 3 different beagle donors. * P<0.05, ** P<0.01, *** P<0.001, **** P<0.0001 via one-way ANOVA with Tukey's post-test.

significantly different (P = 0.08). Notably, however, of the four study subjects with undetectable serum IL-6 at baseline, three patients (75%) were alive at six months compared to four study subjects with detectable serum IL-6 who all died within 6 months of study entry. Two dogs could not be evaluated because of insufficient serum for analysis at the indicated time point.

## Circulating immune phenotype

As shown in Fig 4, we also performed flow phenotyping of circulating immune populations from baseline PBMCs, analyzing the percentage of circulating CD45+ cells for granzyme B

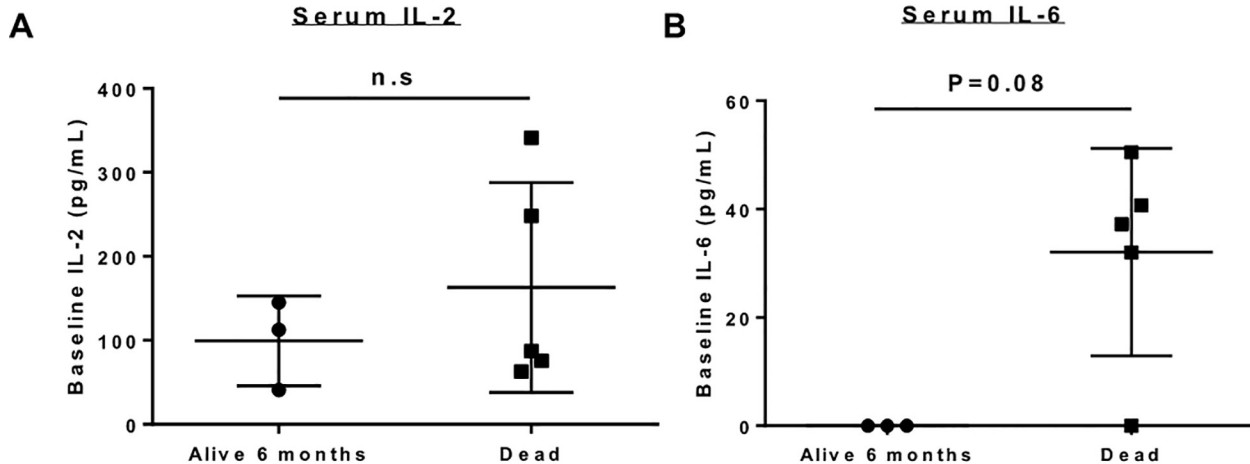

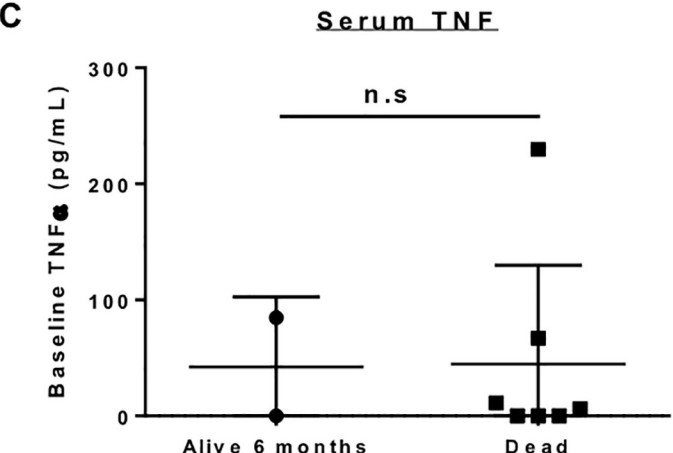

**Fig 3. Baseline serum cytokine expression in trial patients.** Baseline expression of serum cytokines was compared between dogs who were alive at 6 months versus those who died. There was no significant different in serum IL-2 (A) or serum TNF-α (C) between survivors and non-survivors. Baseline serum IL-6 (B) was notably higher in non- survivors (32.1 pg/mL ±19.2) compared to survivors (not detectable), although this difference was not statistically significant (P = 0.08).

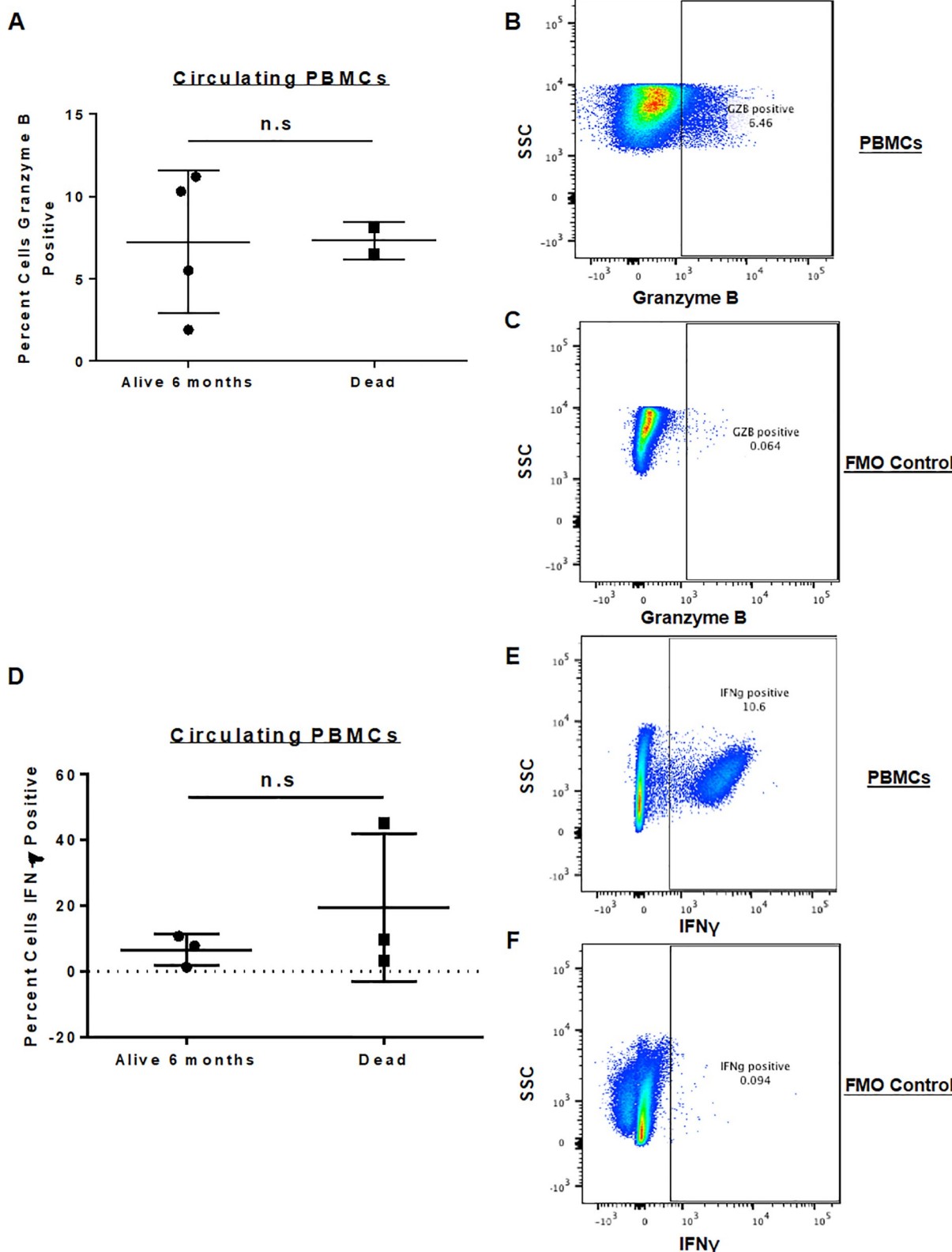

**Fig 4. Expression of Granzyme B and IFN-γ in isolated peripheral blood mononuclear cells.** PBMCs were isolated from dog patients on trial and analyzed via flow cytometry for the expression of activation markers. Baseline PBMC expression between dogs who were alive at 6 months versus those who died showed no significant difference in percent granzyme B positive (A) or percent IFN-γ positive PBMCs (D). Representative flow cytometry staining for granzyme B (B) with FMO control (C) and IFN-γ (E) with FMO control (F) is shown.

and IFN-γ expression at study entry. We have shown previously that granzyme B expression by circulating CD45+ cells is significantly increased in paired patient blood samples after intra-tumoral NK transfer [13]. However, in the current analysis, when looking at baseline circulating immune phenotype (within 7 days of initiation of palliative RT), we observed no significant difference (P > 0.05) in percent positive circulating granzyme B+/CD45+ cells (Fig 4A–4C) between 6-month survivors (mean 7.2 ± 4.8%) versus non-survivors (mean 7.3 ± 1.7%). Similarly, we did not observe significant differences in baseline percent positive IFN-γ+ cells (Fig 4D–4F) with a mean 5.3 ± 5.1% among 6-month survivors versus 19.8 ± 20.5% among non-survivors (P > 0.05).

## Intra-tumoral immune phenotype

As shown in Fig 5, we next evaluated the tumor microenvironment (TME) of OSA in both untreated dogs and study subjects using immunohistochemistry and qRT-PCR. Lymph node tissue from a non-tumor bearing dog (Golden Retriever) and tumor tissue from two soft tissue sarcoma and three OSA patients (all untreated and not study patients) were evaluated for baseline lymphocyte infiltration (Fig 5A). Immunohistochemical analysis of these tumors revealed minimal CD3 infiltration in untreated tumors compared to CD3-rich normal lymph node as a positive control. PCR analysis of PBMCs (Fig 5A) from these same tumor-bearing, but non-study patients showed a statistically greater expression of circulating CD3 transcripts compared to CD8 and NKG2D, and the expression of these transcripts in PBMCs was approximately 100-fold greater than the respective expression in OSA tumor tissue (P < 0.0001). In addition, within OSA tumor tissue (Fig 5A), the expression of the predominantly NK transcript NKG2D was significantly lower than the expression of both CD3 and CD8 (P < 0.05).

We then evaluated the expression of immune-related genes in matched biopsy specimens from eight study dogs where pre- and post-treatment tumor tissue was available before and after palliative RT and intra-tumoral NK. As shown in Fig 5B, there was marked heterogeneity in the expression of immune-related genes after RT plus NK transfer, both within and among patients. For example, two patients (patient 1 and patient 8) showed notable increases in the expression of CD3 and CD8, while two different patients showed greater than 10-fold increases in IL-6 expression. Patient 8 also showed an approximately 40-fold increase in the expression of IDO1 (Fig 5B). We then assessed whether there was any correlation between the changes in gene expression of these immune genes and patient survival. As shown in Fig 5C–5H, we observed no statistically significant differences in changes in expression of genes of interest and survival. Similar to the levels of circulating IL-6, we did observe that study dogs who died within 6 months appeared to have numerically greater (but not statistically different) increases in intra-tumoral expression of IL-6 (Fig 5G), although the two patients with elevated levels may represent outliers in an otherwise small sample size. However, despite the lack of significant differences between survivors and non-survivors, these data reinforce the validity of using these endpoints as hypothesis-generating data for evaluation in future immunotherapy trials.

## The cancer genome atlas

Recent high impact publications using TCGA data have characterized the breadth and depth of genetic and epigenetic abnormalities in multiple human cancers, including sarcomas. Studies of TCGA data have also demonstrated the immune landscape of human malignancy with provocative prognostic implications [19–21]. With these results in mind, we sought to analyze TCGA data from human STS (as data on human OSA are not available) looking for differences in the expression of intra-tumoral immune genes of interest and their relationship with survival (Fig 6). In particular, we focused on genes associated with cytotoxic lymphocyte

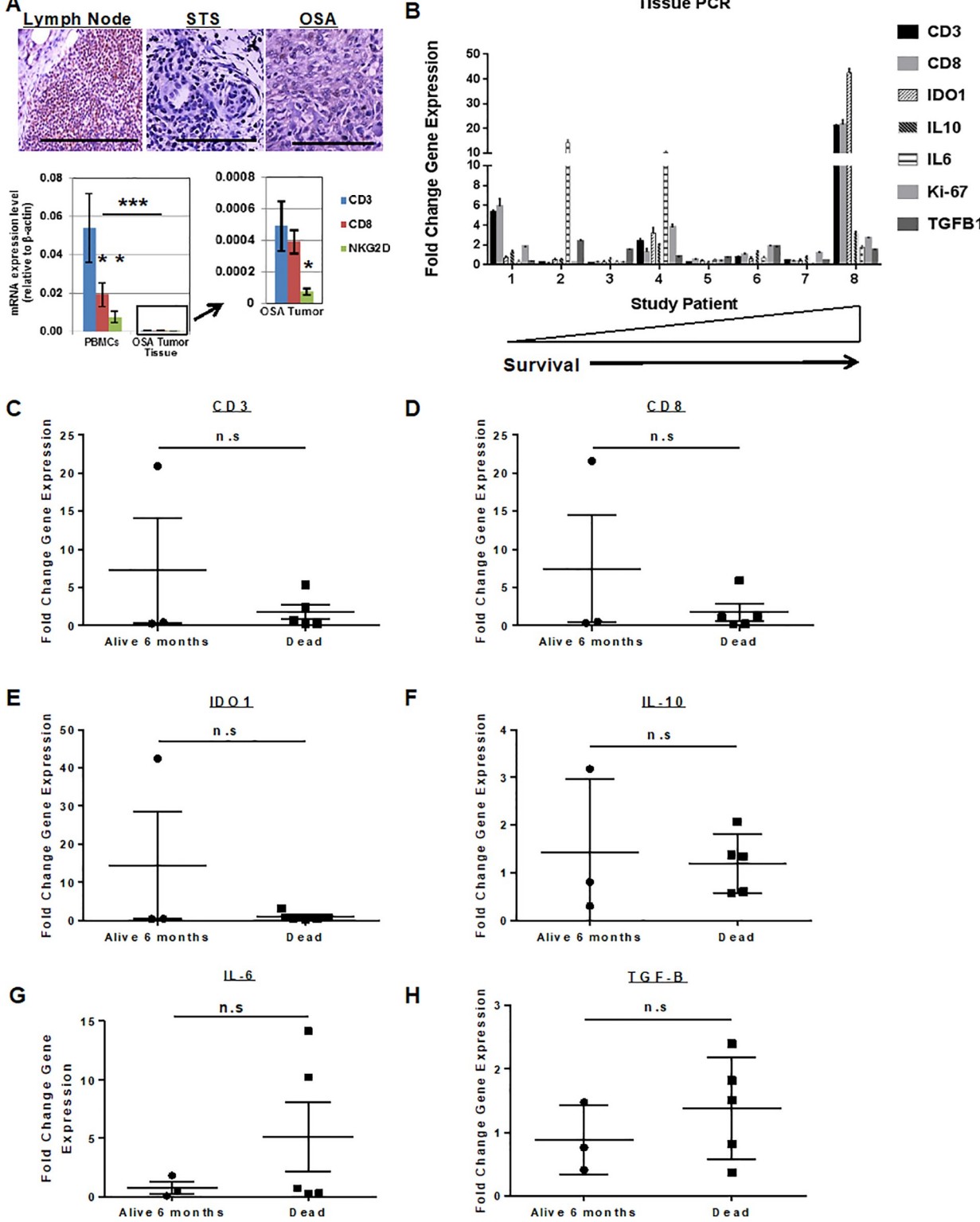

**Fig 5. Intra-Tumoral immune cell infiltration and gene expression in dog sarcomas.** (A) Lymph node from a dog necropsy specimen was compared to tissue from untreated dog soft tissue sarcoma and osteosarcoma cases. Immunohistochemical staining for CD3+ cells revealed high infiltration in normal lymph node (left) versus rare to absent lymphocyte presence in soft tissue sarcoma (middle) or osteosarcoma (right) tumors. Scale bar = 200 µm. RNA analysis by PCR of PBMCs from healthy dogs (A–bottom left) demonstrated greatest expression of CD3 compared to CD8 or NK marker NKG2D. Expression of these genes in PBMCs was approximately 100-fold greater than expression in

osteosarcoma tumor tissue (A–bottom right), and intra-tumoral expression of NKG2D was significantly lower than both CD3 and CD8. Technical replicates from one of 3 experiments are shown. * P<0.05, *** P<0.001 via one-way ANOVA with Tukey's post-test. (B) Tumor tissue from 8 study subjects was obtained at baseline and after palliative RT and intra-tumoral NK and analyzed for change in intra-tumoral gene expression after therapy. qPCR results demonstrated marked heterogeneity in changes in gene expression of key immune-related genes. Interestingly, the patient who lived the longest (17.9 months) showed the greatest fold-change in the expression of CD3, CD8, and IDO1 genes. On univariate analysis, there was no significant difference (P>0.05) in intra-tumoral changes in gene expression and survival for CD3 (C), CD8 (D), IDO1 (E), IL-10 (F), IL-6 (G), or TGF-β (H). Symbols represent fold change in gene expression from before therapy to after using pairwise comparisons from individual treated dogs. Mean fold change with standard deviation is shown. Groups were compared using an unpaired t-test.

phenotype and function (CD3, CD8, Interferon-γ, Granzyme B, Perforin, and IL-2 receptor B/CD122) as well as IL-6 and IL-6 receptor given our serum ELISA and tumor PCR results for IL-6 (Figs 3 and 5). In order to identify potentially meaningful differences in the prognostic effect of these genes of interest, we categorized gene expression levels based on high expression (quartile 4) and low expression (quartile 1).

As shown in Fig 6, we then analyzed survival differences. Notably, we observed that elevated expression in the highest quartile of CD3e (HR 0.51, 95% CI 0.29–0.91, P = 0.02), CD8a (HR 0.45, 95% CI 0.25–0.80, P = 0.006), IFN-γ (HR 0.50, 95% CI 0.28–0.89, P = 0.02), and perforin (PRF1, HR 0.42, 95% CI 0.22–0.80, P = 0.006) in human sarcomas were associated with greater OS. In contrast, although there was a trend for improved survival with greater expression of granzyme B (GZMB) and therefore a lower risk of death with a HR of 0.65, this survival difference was not significantly different (P = 0.13). Interestingly, and somewhat paradoxically given our serum ELISA and tumor tissue PCR results, we observed that higher expression of IL-6 (HR 0.46, 95% CI 0.26–0.82, P = 0.007) and the IL-6 receptor/CD126 (HR 0.46, 95% CI 0.27–0.79, P = 0.004) were both associated with greater OS in human sarcomas (Fig 6G and 6H).

## Discussion

Immunotherapy is rapidly becoming the 4th arm of cancer therapy. Breakthrough advances using inhibitors of PD-1, PD-L1, and CTLA-4 and chimeric antigen receptor (CAR) T cells have clearly identified immunotherapy as the future of clinical oncology. However, despite these exciting advances, more basic and translational research is needed to extend the promise of cancer immunotherapy to greater numbers of patients since significant subsets of patients either do not respond to treatment or develop resistance [22]. A key step in advancing cancer immunotherapy will be the identification of more robust biomarkers of response and resistance [23], and studies in dogs with spontaneous cancers are ideal for this purpose as dogs represent an important link in translating immunotherapy from pre-clinical mouse studies to clinical trials in humans [11].

Dogs with spontaneous tumors have a number of advantages over mice as therapeutic models, including extensive homology between canine and human genomes, comparable genetic complexity and tumor heterogeneity, and similar host-tumor interactions in the TME. Therefore, dog clinical trials are an important mechanism to evaluate novel immunotherapy approaches, especially for NK cellular therapy which has proven challenging to effectively implement in human solid tumors [24]. We have previously reported the feasibility of our first-in-dog clinical trial of intra-tumoral NK in dogs with OSA [13]. In this follow up analysis, we describe the utility of monitoring serum and tumor biomarkers and their correlations with outcome.

Though limited in patient number, our results suggest that there may be a predictive role for pre-treatment serum cytokines and intra-tumoral gene expression. Specifically, this appears most prominent for pre-treatment serum IL-6, in which low expression may correlate

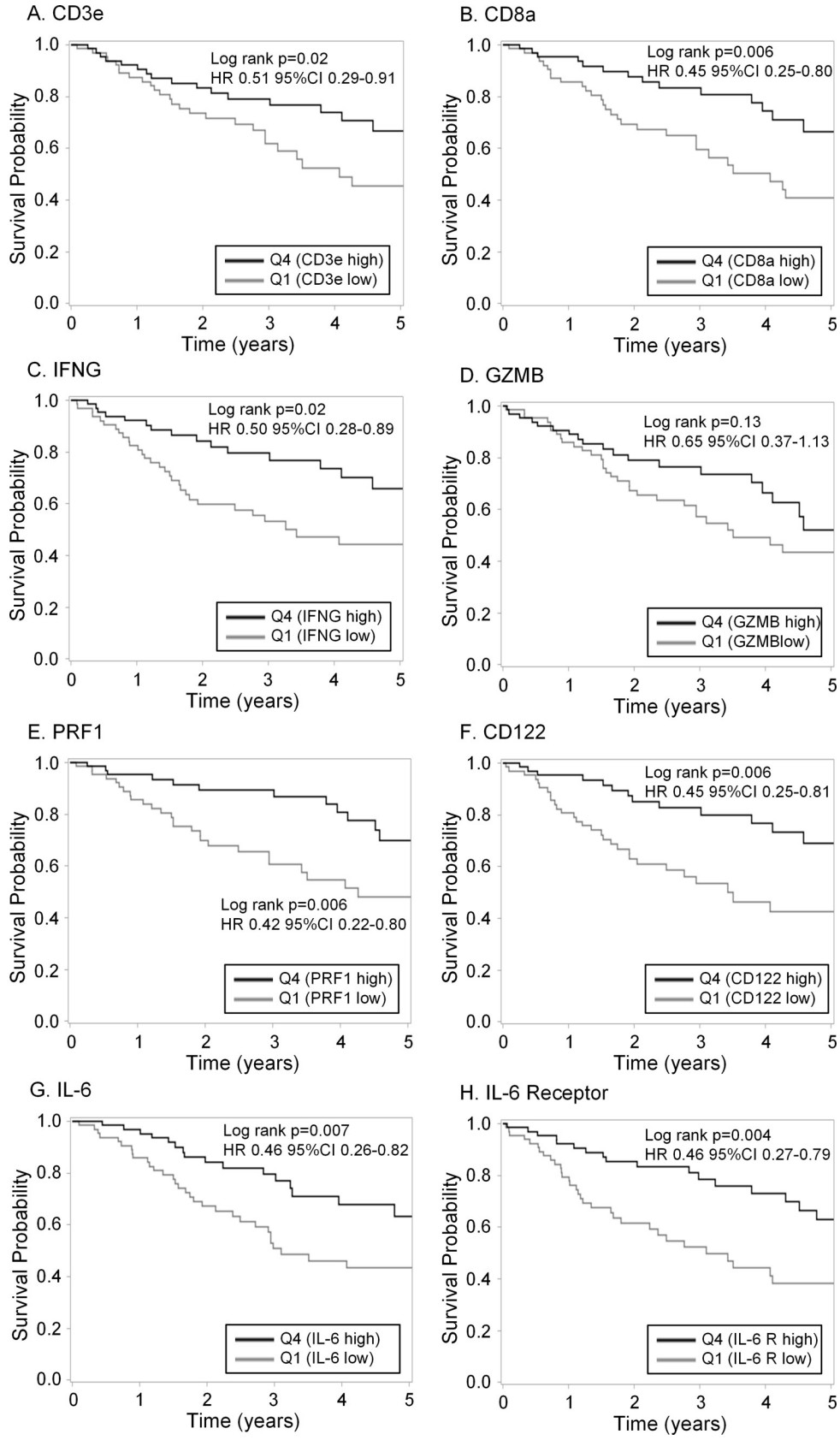

**Fig 6. Intra-tumoral gene expression and survival outcomes in human sarcomas from the cancer genome atlas.**
The Cancer Genome Atlas (TCGA) was queried for gene expression data with immunoregulatory function from
human sarcoma patients with non-metastatic disease. Patients were categorized into high and low expression by
quartiles, and survival differences were analyzed between low expression (quartile 1) and high expression (quartile 4).
Significant survival differences were identified between high and low gene expression for (A) CD3e, (B) CD8a, (C)
IFN-γ (E) PRF1 (perforin), (F) CD122/ interleukin 2 (and interleukin 15) receptor subunit beta, (G) IL-6, and (H) IL-
6R. Notably, and somewhat paradoxically, greater intra-tumoral expression of IL-6 and IL-6R were associated with
superior survival in human sarcomas. Sarcoma patients with higher intra-tumoral GZMB (granzyme B) expression (D)
had longer survival, but this difference was not statistically significant (HR 0.65, 95% CI 0.37–1.13, P = 0.13).

with improved survival and/or response to therapy. This correlation with IL-6 is consistent
with a large body of literature from human studies suggesting that IL-6 adversely impacts can-
cer extent of disease and prognosis [25]. Interestingly, however, analysis of TCGA data sug-
gests that IL-6 and IL-6R expression in the sarcoma TME may favorably impact OS, though
we are unable to differentiate soluble IL-6R on our analysis, which may result in these some-
what paradoxical effects While our study does not answer these critical questions regarding
which immune genes and signatures reliably predict prognosis and response to therapy, it
does lends support to the validity of using the canine model for more detailed immune dissec-
tion. As individual biomarkers often have limited predictive value in larger data sets given the
heterogeneity of human and dog patients, immune signatures and gene clusters may prove to
be more robust prognostic and predictive tools, as was suggested by a recent multispecies anal-
ysis by Scott et al [26].

The search for putative prognostic or predictive biomarkers has been intensely investigated
in cancer therapy, particularly immunotherapy, where only a subset of patients will respond to
treatment. Currently, the most useful immune biomarkers for efficacy of immunotherapy have
been intra-tumoral PD-L1 expression, magnitude of tumor mutational burden [27,28], as well
as the quantity of tumor infiltrating lymphocytes (TILs) pre- and post-therapy [29,30]. These
biomarkers are the subject of much investigation in human trials for their ability to predict
response to immunotherapy. Although no such correlation has yet been made in the canine
population, PD-L1 expression and tumor mutation burden are unlikely to be related to out-
come or response to therapy in non-T-cell based therapies such as NK cellular therapy. Of
note, our limited data do suggest that dog OSA has relatively limited T cell infiltration at base-
line consistent with immunologically "cold" tumors and that intra-tumoral NK transfer can
stimulate infiltration of cytotoxic T lymphocytes [31]. Although additional studies are needed
to validate TILs as a prognostic biomarker in dog OSA, our data demonstrate that both IHC
and PCR can reliably detect these markers in future studies [32]. Similarly, although studies
have identified robust immune infiltrates in dog OSA by next-generation sequencing, the
number of immune cells detectable by IHC is low [26,33].

The intra-tumoral immune phenotype also provides important insight into the TME and
illustrates the correlation between peripheral and intra-tumoral expression. Our data highlight
the difference between the peripheral and intra-tumoral T cell subpopulations, lending further
evidence to the growing body of research on the critical role of TILs in diverse cancer types
[34–38]. Analysis of untreated tumors in our study highlight the relative paucity of immune
infiltration within canine sarcomas, and it is this lack of immune infiltration that promotes the
rationale for immunomodulatory RT with NK adoptive transfer to increase immune infiltra-
tion. The canine patients in this study recapitulate the human finding that subsets of intra-
tumoral immune infiltrates differ from the proportions found in peripheral blood. More so,
changes in these subpopulations indicate that immunotherapy can transform a lymphocyte
negative ("cold") tumor to a lymphocyte positive ("hot") tumor. This is particularly relevant in
the context of tumor MHC-I expression where MHC-I+ tumor cells elicit greater antigen-

specific CD8+ T cell responses and MHC-I loss is a well-established mechanism of tumor immune evasion [39]. Our study did not address tumor MHC-I expression, but prior research has shown loss of MHC-I in human sarcomas [40]. Although this study utilized a single RT regimen consisting of 4 fractions of 9 Gy weekly, evidence suggests that different regimens can alter the immunomodulation elicited. [41,42]. Further studies moving forward could utilize increased radiation dose and decreased fractionation in an attempt to better augment immune infiltration.

Our study is primarily limited by the small number of patients in our trial. However, although limited in number, the rich immunologic and biochemical data generated provides evidence that immune readouts hold promise as potential biomarkers that can be further studied in follow up dog NK trials and validated for use in human NK clinical trials. Future canine immunotherapy trials should incorporate this level of biochemical and tumor monitoring to aid in identifying novel biomarkers with the goal of improving care across species.

## Conclusion

Follow up blood and tissue specimen analysis from our first-in-dog clinical trial of RT and ex vivo expanded autologous NK cell immunotherapy illustrates the feasibility and applicability of utilizing companion dogs for novel cancer therapies. Similar to human studies, analysis of immune markers from canine serum, PBMCs, and tumor tissue is possible and provides valuable immune monitoring data for biomarker analysis. These results reinforce the value of canine immunotherapy trials to speed translation of novel immunotherapy approaches with particular emphasis on identifying novel predictive and prognostic biomarkers that may be beneficial to both dog and human patients.

## Supporting information

**S1 Table. List of canine specific primers.**
(DOC)

**S2 Table. List of canine specific antibodies.**
(DOC)

## Acknowledgments

This work was supported in part by National Institute for Health/National Cancer Institute grant R01 CA189209 (WJM) and U01 CA224166–01 (RJC, RBR). Additional funding was provided by the Society for Surgical Oncology Foundation (RJC), the Sarcoma Foundation of America (RJC), and the University of California Coordinating Committee for Cancer Control CRR-13-201,404 (RJC). We thank Jonathan Van Dyke and Bridget McLaughlin from the Flow Cytometry Core Facility at the University of California Davis Comprehensive Cancer Center, which also receives funding from the National Cancer Institute, for support with the acquisition and evaluation of patient samples. The authors are also grateful to Teri Guerrero, Heather Schrader, Frank O'Daniel, and the clinicians on the UC Davis VMTH Oncology Clinical Trials Service for veterinary clinical trials support.

## Author Contributions

**Conceptualization:** Sean J. Judge, Ian R. Sturgill, Jaime F. Modiano, William J. Murphy, Michael S. Kent, Robert J. Canter.

**Data curation:** Sean J. Judge, Mio Yanagisawa, Ian R. Sturgill, Alicia A. Gingrich, Arta M. Monjazeb, Michael S. Kent, Robert J. Canter.

**Formal analysis:** Sean J. Judge, Ian R. Sturgill, Sarah B. Bateni, William J. Murphy, Michael S. Kent, Robert J. Canter.

**Funding acquisition:** William J. Murphy, Robert J. Canter.

**Investigation:** Sean J. Judge, Mio Yanagisawa, Ian R. Sturgill, Sarah B. Bateni, Jennifer A. Foltz, Dean A. Lee, William T. N. Culp, Robert B. Rebhun, William J. Murphy, Michael S. Kent, Robert J. Canter.

**Methodology:** Sarah B. Bateni, Alicia A. Gingrich, Arta M. Monjazeb, William J. Murphy, Robert J. Canter.

**Project administration:** Robert J. Canter.

**Resources:** Jennifer A. Foltz, Dean A. Lee, William J. Murphy, Robert J. Canter.

**Supervision:** William J. Murphy, Robert J. Canter.

**Validation:** Jennifer A. Foltz, Dean A. Lee, William J. Murphy.

**Writing – original draft:** Sean J. Judge, Mio Yanagisawa, Ian R. Sturgill, Robert J. Canter.

**Writing – review & editing:** Sean J. Judge, Jaime F. Modiano, Arta M. Monjazeb, William T. N. Culp, Robert B. Rebhun, William J. Murphy, Michael S. Kent, Robert J. Canter.

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
