## [Decision Letter · Decision Letter 0]

17 Jan 2020

PONE-D-19-28856

Blood and Tissue Biomarker Analysis in Dogs with Osteosarcoma Treated with Palliative Radiation and Intra-Tumoral Autologous Natural Killer Cell Transfer

PLOS ONE

Dear Dr. Canter,

Thank you for submitting your manuscript to PLOS ONE. After careful consideration, we feel that it has merit but does not fully meet PLOS ONE’s publication criteria as it currently stands. Therefore, we invite you to submit a revised version of the manuscript that addresses the points raised during the review process.

Please response and make necessary changes to your text as per reviewer's comments. Also I have following question and suggestion.

Could you please explain if any form of allogeneic transplantation influences experimental outcomes?

We would appreciate receiving your revised manuscript by January 31st. To enhance the reproducibility of your results, we recommend that if applicable you deposit your laboratory protocols in protocols.io, where a protocol can be assigned its own identifier (DOI) such that it can be cited independently in the future. For instructions see: http://journals.plos.org/plosone/s/submission-guidelines#loc-laboratory-protocols

We look forward to receiving your revised manuscript.

Kind regards,

Nupur Gangopadhyay, B.V.Sc, M.V.Sc.,Ph.D.

Academic Editor

PLOS ONE

Journal Requirements:

2. If applicable, please include in your Methods section the gene expression datasets (or links to those) used from TCGA for analysis.

3. In your Methods section, please give the sources of any cell lines used in your study (K562, CTAC).

Reviewers' comments:

Reviewer's Responses to Questions

**Comments to the Author**

1. Is the manuscript technically sound, and do the data support the conclusions?

Reviewer #1: Yes

2. Has the statistical analysis been performed appropriately and rigorously? 

Reviewer #1: Yes

3. Have the authors made all data underlying the findings in their manuscript fully available?

Reviewer #1: Yes

4. Is the manuscript presented in an intelligible fashion and written in standard English?

Reviewer #1: Yes

5. Review Comments to the Author

Reviewer #1: Robert Cantor and colleagues have previously shown that radiation therapy sensitizes tumor cells, including osteosarcoma to NK cell-mediated cytoctoxicity. As a follow-up to this discovery, they conducted a clinical trial of palliative RT plus intra-tumoral autologous NK transfer in dogs with non-metastatic osteosacrcoma, whose owners elected not to pursue amputation or cytotoxic chemotherapy. As part of the trial, the investigators collected serial blood specimens and tumor tissue pre- and post-therapy. This samples have been analyzed for serum levels of cytokines, immunophenotyping of the peripheral blood mononuclear cells and gene expression in the tumor tissue for this report. Ten dogs were treated, of which only four are alive. Four died from tumor progression, one died from duodenal perforation (secondary to NSAID use) and another was euthanized with pathological fracture post-RT and local recurrence. Key observations include:

NK cells could be isolated after depleting CD5+ cells and expanded ex vivo from PBMC of trial subjects. Functional cytotoxity was demonstrated in these CD3-NKp46+ cells using canine thyroid adenocarcinoma cells as a target.

Serum cytokines (IL-2, TNFa and IL-6) were not statistically different amongst study subjects. However, compared to subjects with disease progression, serum IL-6 was mostly undetectable amongst survivors without tumor progression.

Granzyme B and IFNg-positive cells were not statistically increased in the peripheral blood amongst subjects that were alive at 6 months versus those who died within 6 months.

CD3+, CD8+ and NKG2D-expressing cells were minimal to absent in tumor tissues from sarcoma and osteosarcoma tissues. Intra-tumoral NKG2D-expressing cells were significantly lower than CD3-expressing or CD8-expressing cells in tumor tissues. On univariate analysis, there was no significant difference in CD3, CD8, IDO1, IL-10, IL-6 or TGFß gene expression. However, the subject that lived the longest (17.9 m) showed the greatest increase in expression of CD3, CD8 and IDO1 RNA.

TCGA analysis showed a significant survival difference between patients with high and low gene expression of CD3e, CD8a, IFNg, PRF1 and CD122. In contrast to data from canine trial, patients with higher IL6 and IL6R had a better survival in humans.

In conclusion, although this study is limited by the number of patients, it suggests that pre-treatment cytokine (IL6) and intra-tumoral gene expression of cytotoxic cells may be predictive of survival and tumor response in canine patients with osteosarcoma. The authors should be congratulated to conduct immunological studies in canine patients.

However, a few points need to be discussed before publication.

Limitation of the study design. The weekly 9 Gy x 4 fraction regimen may not be ideal for an immunogenic activation. Please discuss how the next studies would be done so as to harness the power of immunogenic RT fractionation and dose.

Did the OSA tumors express HLA?

If NKG2D-expressing cells were minimal to absent, please discuss how NK cell therapy would be effective.

In the TCGA analysis, the authors could examine the expression of soluble IL6R, if possible. Because that might be confounding towards the paradoxical results in TCGA versus canine patients.

6. PLOS authors have the option to publish the peer review history of their article (what does this mean?). If published, this will include your full peer review and any attached files.

Reviewer #1: Yes: Chandan Guha, MBBS, PhD

---

## [Author Response · Author response to Decision Letter 0]

31 Jan 2020

Nupur Gangopadhyay, B.V.Sc, M.V.Sc.,Ph.D. January 31, 2020

Academic Editor

PLOS ONE

RE: PONE-D-19-28856

 Blood and Tissue Biomarker Analysis in Dogs with Osteosarcoma Treated with Palliative 

 Radiation and Intra-Tumoral Autologous Natural Killer Cell Transfer

Dear Dr. Gangopadhyay and PLOS ONE reviewers,

Thank you for your comprehensive review of our manuscript. We appreciate the editor’s and reviewer’s comments, and we have attempted to address these comments with a revised version of our manuscript. We hope that you will now consider our manuscript suitable for publication. Please see the following point-by-point responses to the individual comments of the editor and reviewers.

Comments to the Editor

Could you please explain if any form of allogeneic transplantation influences experimental outcomes? 

This is an important question as allogeneic immune interactions can have significant effects on the outcomes of cellular therapies both in terms of anti-tumor effects as well as toxicities. In human clinical trials, a strong trend in the field is to pursue allogeneic NK transfer as graft-versus-host disease has not been associated with pure NK cell populations (GVHD is primarily mediated by T cells and significant pre-clinical evidence implicates a protective effect of NK cells in GVHD), while allogeneic NK cells have been associated with improved anti-tumor effects (so-called graft-versus-tumor [GVT] or graft-versus-leukemia [GVL]) in human studies. In our study, since it was a first-in-dog clinical trial and since dog NK cells are not as well characterized as human or mouse NK cells, we intentionally decided to use autologous canine NK cells as an extra precaution to mitigate toxicity given the potential for unexpected toxicities to occur in clinical trials (e.g., if our NK product was not as T-cell depleted as our background data suggested). Therefore, in the dog clinical trial patients we are reporting in this study, we can conclude that there is no effect of allogeneic interactions on the results. However, given that our data suggest that NK transfer in dogs appears to be safe (based on the data available to date) and given our ongoing studies improving the characterization of dog NK cells, we are pursuing allogeneic NK transfer in our follow-up studies since it is reasonable to hypothesize superior anti-tumor effects in these cases. However, key questions will be how allogeneic NK transfer impacts engraftment, activation status, longevity on the transferred NK cells (as well as how much conditioning is necessary to prevent rejection by the host’s immune cells). These are all important comments which relate to the editor’s question, and we intend to address them in follow-up studies. 

A final point is that our current technique to expand and activate autologous canine NK cells is completed through co-culture with a human (and xenogeneic) K562.Cl9 cell line that has been engineered to expressed human membrane bound IL-21 and 4-1BB ligand. This cell line has been well validated for the expansion of human (and now canine) NK cells and undergoes irradiation prior to culturing to prevent expansion and engraftment following adoptive cell transfer. As human and canine NK cells respond similarly to the transfected K562.Cl9 feeder cell line, there does not appear to be any demonstrable xenogeneic effects identified in the canine NK cells as compared to human NK cells. These points have been expanded in the methods section where we describe the clinical trial (page 4) as well as the discussion section, and we thank the editor for this question. 

Comments to the Author

5. Review Comments to the Author

Limitation of the study design. The weekly 9 Gy x 4 fraction regimen may not be ideal for an immunogenic activation. Please discuss how the next studies would be done so as to harness the power of immunogenic RT fractionation and dose.

We appreciate the reviewer’s comment and attention to our study design. This is an important area of investigation in the intersecting fields of radiotherapy and immunotherapy. The immunomodulatory effects of radiotherapy, particularly local immune effects on the tumor microenvironment, have been established in preclinical models and include induction of immunogenic cell death, release of antigens for T cell priming, increased T cell homing to tumor sites, shift in the polarization of tumor associated macrophages, and reduction of immunosuppressive stromal cells in the tumor microenvironment, among others. However, radiotherapy can also augment immunosuppressive pathways, including induction of TGF-B and PD-L1, among others. More recent studies in humans suggest that hypo-fractionated radiation schedules produce very different biologic effects than traditional conventionally fractionated radiation, and few studies have addressed these variables in dogs undergoing RT. Especially in dogs, limited preclinical or clinical data are available to guide the selection of radiation dose, fractionation and site in order to optimally synergize with immunotherapy. Therefore, we completely agree with the reviewer that this is a key question for follow-up studies. Since palliative RT protocols are fairly fixed in clinical medicine (human as well as veterinary), and since studies in dogs with cancer nevertheless maintain a focus on the clinical care of the animals, we do not anticipate that it will be feasible to rigorously vary RT doses and fractionation schedules to comprehensively assess the optimal RT strategy for immunogenic activation (which may nevertheless vary by tumor type and target organ like lungs, liver, or soft tissue (McGee et al., 2018. PMID: 29891204). However, moving forward, we are prospectively collecting immune cells and serum from blood and tumor tissue at multiple time points from dog cancer patients undergoing RT with and without immunotherapy (follow up dog NK clinical trials) to more comprehensively assess the effects of RT on immune phenotype and function as the reviewer suggest. Ultimately, we agree that this will be an important area of research in dogs as it is in humans. However, we also anticipate significant heterogeneity in the immune effects of RT which will vary by irradiated site (e.g. brain and bone are more immunosuppressive micro-environments than the lungs) as well as breed, RT dose, schedule, etc…in which case, this concern may remain an ongoing limitation of radio-immunotherapy studies. These points have been added to our discussion section (page 13-14) and additional references have been added to our discussion section (Ref 44-46: Immunobiology of Radiotherapy: New Paradigms; Unlocking the combination: potential of radiation-induced antitumor responses with immunotherapy; Stereotactic ablative radiotherapy induces systemic difference sin peripheral blood immunophenotype dependent on irradiated site). 

Did the OSA tumors express HLA?

Our analysis of the OSA clinical samples did not include expression of canine MHC-I or II, in part because dog leukocyte antigen (DLA) is highly polymorphic (as it is in humans) and also because robust reagents to evaluate DLA phenotype are limited at this time (JL Wagner, 2003. PMID: 12692158). However, we completely agree with the reviewer that this is an important area for future study in dog immunotherapy as these haplotypes have clearly been shown to influence CD8 T cell and NK cell effector function in human immunotherapy studies. As we discuss above in our response to the editor’s comment, this is a significant focus of our ongoing investigations into methods to optimize dog NK characterization as well as dog NK adoptive transfer. These points have been added to our discussion section on page 13. 

If NKG2D-expressing cells were minimal to absent, please discuss how NK cell therapy would be effective.

We agree with the reviewer that this is an important to emphasize, and these results suggest that strategies to stimulate endogenous anti-tumor immunity from cytotoxic NKG2D-expressing cells like NK cells and bystander T cells may be less likely to succeed. However, we hypothesize that this is where improving NK adoptive transfer techniques could be promising since exogenous delivery of cytotoxic NKG2D-expressing NK and T cells can overcome some of the limitations of “cold” tumor with low baseline immune infiltrate. In our data, the bone and soft tissue sarcoma tumors of untreated dogs showed low expression of NKG2D, in addition to other markers of immune effector cells (CD3, CD8). This highlights the relative paucity of immune cell within canine sarcomas and underlies the rationale for a therapy that induces immunomodulation (RT) and aims to increase infiltration of an effector cell population (NK cells). We have modified the manuscript to clarify these points. 

In the TCGA analysis, the authors could examine the expression of soluble IL6R, if possible. Because that might be confounding towards the paradoxical results in TCGA versus canine patients.

We appreciate this additional insight and suggestion from the reviewer, as he or she has identified a notable paradox of our data which also intrigued us. The implication from our primary dog data is that over-expression of IL-6 is a negative prognostic factor in dog osteosarcoma while in the TCGA data overexpression of both IL-6 and the IL-6 receptor was associated with superior survival in soft tissue sarcoma patients. Therefore, we certainly agree that further analysis of these results is warranted to reconcile the differences between dog and human which may be species specific, tumor specific, or related to other factors like immune responses and/or therapy (e.g. use of RT in our dog patients). Unfortunately, the TCGA does not delineate membrane-bound versus soluble IL-6R expression in the dataset, although this clearly would be a relevant area to start to address this paradox. The IL-6R can be produced from an alternative splicing event or via proteolytic cleavage at the protein level. To detect RNA level differences from the TCGA, a unique identifier is needed to detect the soluble form of the transcript, and this is not currently available through the TCGA system. Although IL-6 is considered a proto-typical pro-inflammatory cytokine that directs and amplifies the acute phase response to tissue damage and inflammation, IL-6 is also viewed as an immunosuppressive cytokine because of its association with multiple regulatory or suppressive immune-cell populations, such as myeloid-derived suppressive cells, tolerogenic dendritic cells, and tumor-associated macrophages. Further evidence has been drawn from studies correlating circulating IL-6 levels with poor prognosis of various cancers, especially breast cancer. However, data have also demonstrated a role for IL-6 in boosting T cell trafficking to tumors, where they can become activated and cytotoxic (Romano et al, 1997. PMID: 9075932). These points, including discussion on soluble IL-6R, has been added to our revised manuscript on page 12. 

Journal Requirements:

1. Please ensure that your manuscript meets PLOS ONE's style requirements, including those for file naming. The PLOS ONE style templates can be found athttp://www.plosone.org/attachments/PLOSOne_formatting_sample_main_body.pdf andhttp://www.plosone.org/attachments/PLOSOne_formatting_sample_title_authors_affiliations.pdf

We have addressed this in our revised manuscript.

2. If applicable, please include in your Methods section the gene expression datasets (or links to those) used from TCGA for analysis.

This has also been addressed in the revised manuscript.

3. In your Methods section, please give the sources of any cell lines used in your study (K562, CTAC).

We have added this information in the revised manuscript. 

4. Please include captions for your Supporting Information files at the end of your manuscript, and update any in-text citations to match accordingly. Please see our Supporting Information guidelines for more information:http://journals.plos.org/plosone/s/supporting-information.

This information has been included in the revised manuscript.

• A rebuttal letter that responds to each point raised by the academic editor and reviewer(s). This letter should be uploaded as separate file and labeled 'Response to Reviewers'.

• A marked-up copy of your manuscript that highlights changes made to the original version. This file should be uploaded as separate file and labeled 'Revised Manuscript with Track Changes'.

• An unmarked version of your revised paper without tracked changes. This file should be uploaded as separate file and labeled 'Manuscript'.

These items are included with our manuscript resubmission.

Thank you again for your detailed review of our manuscript. We have attempted to address all points raised to improve our manuscript. We hope our revised manuscript is now suitable for publication in PLOS ONE.

Sincerely,

Robert J. Canter, MD

Professor of Surgery

Sarcoma Services, Surgical Oncology

University of California at Davis

---

## [Editor Report · Decision Letter 1]

5 Feb 2020

Blood and Tissue Biomarker Analysis in Dogs with Osteosarcoma Treated with Palliative Radiation and Intra-Tumoral Autologous Natural Killer Cell Transfer

PONE-D-19-28856R1

Dear Dr. Canter,

We are pleased to inform you that your manuscript has been judged scientifically suitable for publication and will be formally accepted for publication once it complies with all outstanding technical requirements.

With kind regards,

Nupur Gangopadhyay, B.V.Sc, M.V.Sc.,Ph.D.

Academic Editor

PLOS ONE
---

## [Editor Report · Acceptance letter]

13 Feb 2020

PONE-D-19-28856R1 

Blood and Tissue Biomarker Analysis in Dogs with Osteosarcoma Treated with Palliative Radiation and Intra-Tumoral Autologous Natural Killer Cell Transfer 

Dear Dr. Canter:

I am pleased to inform you that your manuscript has been deemed suitable for publication in PLOS ONE. Congratulations! Your manuscript is now with our production department. 

With kind regards,

on behalf of

Dr Nupur Gangopadhyay 

Academic Editor

PLOS ONE